# Current Concepts of the Applications and Treatment Implications of Drug-Induced Sleep Endoscopy for the Management of Obstructive Sleep Apnoea

**DOI:** 10.3390/diagnostics15202614

**Published:** 2025-10-16

**Authors:** Chi Ching Joan Wan, Yiu Yan Leung

**Affiliations:** Oral and Maxillofacial Surgery, Faculty of Dentistry, The University of Hong Kong, Hong Kong, China; wccjoan@gmail.com

**Keywords:** airway management, drug-induced sleep endoscopy, obstructive sleep apnoea, sleep-disordered breathing, surgery therapy

## Abstract

Obstructive sleep apnoea (OSA) is a complex health condition associated with significant health risks and diminished quality of life. Despite continuous positive airway pressure (CPAP) being the gold standard treatment for years, its poor adherence is well documented. With the emergence of drug-induced sleep endoscopy (DISE) and phenotypic approach to OSA, traditional surgical and non-surgical treatment pathways have been improved to allow personalised treatment and minimising suboptimal treatment to patients demonstrating various upper airway obstruction of OSA endotypes. Sedation protocol propofol, midazolam and dexmedetomidine have been suggested. The VOTE classification for documenting DISE findings have been proposed to unify results across studies. DISE plays an invaluable role in offering insights on treatment successes for positive airway pressure (PAP) therapy, mandibular advancement device (MAD) therapy, positional therapy, and surgical interventions including palatal surgeries, tongue base surgeries, upper airway stimulation (UAS) surgery and maxillomandibular advancement (MMA). This review aims at consolidating current evidence on DISE protocols, indications, and treatment implications to improve therapeutic success in OSA management.

## 1. Introduction

Patients suffering from obstructive sleep apnoea (OSA) experience subjective nuisances [1,2,3] and are exposed to a range of health risks [4,5,6,7,8,9]. Sleep fragmentation adversely affects several crucial brain functions, including emotional stability [10], memory consolidation [11], glymphatic clearance [12], and overall cognitive performance [13].

An impaired anatomy, characterised by a narrow or collapsible upper airway, is the primary cause of OSA in approximately 80% of patients. The remaining 20% of cases are due to non-anatomical phenotypes, such as impaired airway dilator muscle control, low respiratory arousal threshold, and high loop gain [14]. Although continuous positive airway pressure (CPAP) remains the golden standard for the treatment of OSA due to its proven efficacy, its compliance was known to be low [15]. Furthermore, CPAP does not provide a curative effect when the device is not in use, prompting patients to seek alternative treatment options that may allow them to eliminate the need for long-term CPAP usage. Hence, a spectrum of treatment alternatives, including surgical interventions, is available to accommodate individualised needs. Non-surgical options include weight loss, positional therapy, oral appliance therapy, myofunctional therapy, and other pharmacological therapies.

Bariatric surgeries remain an option for overweight and obese patients. For patients with an acceptable body mass index (BMI) [16], upper airway surgeries can be performed targeting specific anatomical sites, including the nose, soft tissues such as the oropharynx and tongue base, and the skeletal framework. Maxillomandibular advancement (MMA) remains the most effective surgical intervention, achieving an 85.5% success rate, defined as a 50% reduction in the apnoea–hypopnea index (AHI) with a postoperative AHI below 20, and a 38.5% cure rate, defined as an AHI under 5 [17]. It will thus be important to understand how we can further improve the outcomes for MMA from case selection, to planning, to execution. The traditional surgical algorithm published by Riley et al. separated the stages to two phases, with MMA performed as a phase II surgery upon failure of phase I soft tissues surgeries. The group has updated the surgical algorithm into a more multimodal, multidirectional fashion, most importantly highlighting the importance of drug-induced sleep endoscopy (DISE) in treatment decisions. The literature has shown that soft tissue surgeries are not effective in patients demonstrating complete concentric collapse (CCC) or lateral pharyngeal wall (LPW) collapse [18,19]; hence, the MMA could be considered at an earlier stage for these patients. Recent publications have emphasised contemporary approaches to OSA, including myofunctional therapy and orofacial exercises, along with emerging perspectives gained through DISE-assisted diagnostic techniques [20,21,22].

DISE is commonly employed as a diagnostic tool to evaluate the level, severity, and pattern of upper airway obstruction, aiding in the selection of appropriate surgical interventions. The VOTE classification [23], introduced in 2011, is among the most widely adopted systems for categorising DISE findings, with VOTE representing the velum, oropharynx, tongue base, and epiglottis. Most studies exploring DISE-guided interventions have focused on evaluating outcomes of oral appliance therapy and soft tissue surgeries. However, there is no consensus on how DISE findings should inform evidence-based treatment planning for MMA to optimise therapeutic outcomes. This comprehensive review seeks to consolidate current knowledge to enhance treatment planning when utilising DISE as a diagnostic tool.

## 2. Methods

This paper presents a narrative review of the development of DISE, explores the recommended protocols and documentation, and discusses the treatment implications derived from DISE findings. To identify the related articles, a comprehensive search was conducted via PubMed, Cochrane Library, and Google Scholar from 2000 to 2025. Keywords used during the search included “drug-induced sleep endoscopy”, “obstructive sleep apnoea”, “treatment”, “surgery”, and “maxillomandibular advancement”. The focus in this review was on adults; hence, papers pertaining solely to paediatric populations were omitted.

## 3. Drug-Induced Sleep Endoscopy

Currently, conducting sleep endoscopy prior to upper airway surgeries for OSA is widely regarded as a clinical standard. This is supported by multiple studies demonstrating a poor correlation between awake nasoendoscopy (NE) [24,25,26] or lateral cephalometry [27,28] and sleep nasoendoscopy, even when manoeuvres like the Müller manoeuvres are used during awake NE to assess upper airway collapsibility. Although natural sleep endoscopy (NSE) ideally allows direct observation of pharyngeal airway collapsibility [29], its labour-intensive nature and time inefficiency limit its practicality as a diagnostic tool. Research indicates that DISE shows similar findings in the upper airway collapsibility as observed in NSE [30,31]. However, one study reported that, despite good agreement between the two methods, DISE identified a greater number of sites with partial or complete obstruction, consequently indicating more sites requiring intervention compared to NSE [30]. It is essential for clinicians to understand the existing DISE protocols, reporting frameworks, and associated treatment outcomes to optimise clinical decision-making. Multiple publications have sought to standardise the procedure, with the 2017 Update of the European position paper being among the most frequently cited [32].

### 3.1. Indications and Contraindications [32,33,34,35,36,37]

DISE is predominantly utilised for patients with OSA. However, its application may also be considered in snorers, should the clinicians deem the additional information may optimise treatment [32]. While DISE serves as a valuable diagnostic tool for evaluating surgical options in OSA, it carries anaesthetic risks, which are heightened in patients with OSA due to their increased susceptibility to anaesthesia-related complications including airway compromise. Careful selection of suitable candidates is essential when scheduling patients for DISE. All patients should undergo a sleep study before proceeding with DISE.

#### 3.1.1. Indications

DISE is indicated in patients who are seeking alternative treatment options to positive airway pressure (PAP) therapy, including oral appliance therapy (OAT), positional therapy (PT), upper airway surgery, or a combination of options. In addition, DISE serves as a tool for titration of OAT and for identifying potential causes of treatment failure who exhibit partial or non-responsiveness to PAP therapy, OAT, or surgical interventions. 

#### 3.1.2. Contraindications

Absolute contraindications to DISE include known allergies to its sedative agents, pregnancy and American Society of Anaesthesiologists (ASA) class 4, due to significant risks posed to the patients. Relative contraindications include morbid obesity, as these patients tend to experience lower success rates with non-PAP treatment alternatives.

### 3.2. Before DISE

#### 3.2.1. Sleep Test

Prior to conducting DISE, an appropriate sleep study should be performed. Although a level I overnight polysomnography (PSG) is considered the ideal diagnostic approach, it is often not feasible for all patients due to resource limitations. Consequently, the use of home sleep tests has been explored, but clinicians should be aware that they may underestimate the severity of OSA, especially in mild cases [38,39]. The American Academy of Sleep Medicine recommends utilising either PSG (levels I and II), or a home sleep test (level III) for diagnosing OSA in uncomplicated adult patients [40]. PSG should be performed in patients presenting with significant cardiorespiratory comorbidities, awake hypoventilation, chronic opioid use, a prior history of stroke, or severe insomnia. It is advisable for surgical cases to have PSG [41] to better categorise OSA and before invasive treatment. The usage of the same level and scoring criteria of sleep study while evaluating treatment efficacy is also vital.

#### 3.2.2. Examinations [41,42]

A comprehensive physical examination should be conducted prior to DISE. The primary objective of the examination is to identify potential etiologies of upper airway obstruction and to serve as a preliminary assessment of the patient’s suitability for various treatment modalities. Given the multidisciplinary nature of OSA management, it is not uncommon for various specialties to have different examination focuses. However, the assessment can broadly be categorised into general, dentofacial, and nasal components, followed by a nasoendoscopy. General parameters including body weight, height, neck circumference, and blood pressure should be documented. From a dentofacial perspective, evaluation should include analysis of skeletal classification and craniofacial proportions, temporomandibular joint function, and dentition, with particular attention to any missing teeth, the occlusion, and the inter-arch relationships across anteroposterior (AP), vertical, and transverse dimensions. Additional assessments include the modified Mallampati score, tonsillar grading, and presence of tongue scalloping. Some practitioners may also incorporate myofunctional examinations [43,44,45], although it is not well documented in available guidelines [46]. NE is performed in the erect and supine positions, initiated at the nasal cavity, and proceeds through the entire upper airway, terminating at the level of the epiglottis, with the objective of identifying potential sites of obstruction. The examination may be complemented by performing manoeuvres such as Muller’s manoeuvre [47,48] to add more diagnostic value. Jaw thrust is also performed to observe the change in the airway dimensions.

#### 3.2.3. Pre-Anaesthetic Consultation

Pre-anaesthetic assessment should be conducted prior to DISE. The anaesthetist would assess the patient’s suitability for the DISE and explain the procedure and the risks involved. Anaesthetic risks involved in DISE include potential allergic reaction to the sedative drug and severe hypoxic events resulting from oversedation or laryngospasm, which may necessitate abortion of the procedure and airway rescue interventions.

### 3.3. During DISE (Figure 1)

DISE offers insight into the nature and extent of upper airway collapse during sleep under sedation. The sedation protocol should be able to simulate the patient’s natural sleep to ensure diagnostic accuracy. While ideally all stages would be observed, this is not practical. Clinicians should balance the need for sufficient diagnostic data with the increased risks associated with prolonged duration of DISE. It is not uncommon to have desaturations during DISE. The clinician should be familiar with the desaturation figures, particularly the lowest oxygen saturation, of the sleep study obtained from the patient’s natural sleep. This enables appropriate intervention, such as the jaw thrust manoeuvre, in the event of significant or prolonged desaturation during induced sleep. Adequate support, including trained personnel, appropriate equipment, and a well-prepared environment, is essential not only for successful execution of DISE, but also for the management of potential airway emergencies during the examination. 

**Figure 1 diagnostics-15-02614-f001:**
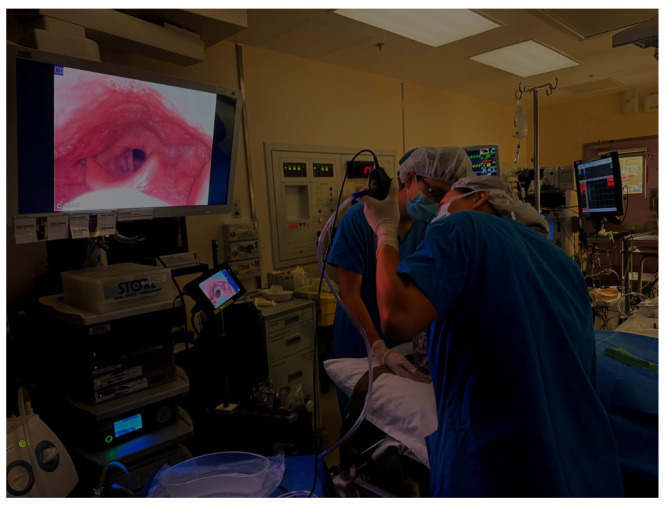
A surgeon performing DISE in a dimmed environment.

#### 3.3.1. Manpower

The team should include (1) the primary clinician(s) performing the endoscopy, typically an otorhinolaryngologist and/or an oral and maxillofacial surgeon; (2) an anaesthetist (whenever feasible, a dedicated one); (3) an assistant to the clinician in performing additional manoeuvres; and (4) an assistant to the anaesthetist. Members should work in accordance with region-specific sedation safety guidelines.

#### 3.3.2. Equipment and Set-Up

To facilitate safe and effective sedation, the availability of standard equipment is essential. This includes an infusion pump, monitoring devices such as pulse oximetry, electrocardiogram, and blood pressure monitor, as well as a bag-valve mask, airway emergency cart, and preferably bispectral index (BIS) monitoring. For endoscopic procedures, essential tools include a flexible endoscope, anti-fog solution, and suction apparatus.

The patient should be positioned comfortably in bed with a pillow, replicating typical sleeping conditions as closely as possible. The operator should have simultaneous visibility of both the monitoring and endoscopy screens [49].

#### 3.3.3. Sedative Drugs 

The mostly used sedative drugs for DISE are propofol, midazolam, and dexmedetomidine. The discussion on the advantages and disadvantages is mainly related to their ability to mimic sleep, the time to onset, ease of titration, their respiratory depressive effects, influence on muscle tone, and cardiovascular effects (Table 1). 

Less commonly used drugs include ketamine and diazepam. Remifentanil or dexmedetomidine may be administered in combination with propofol or dexmedetomidine, aiming to achieve synergistic effect and reduce the required dose of the primary sedative to mitigate associated effects [54]. However, a study suggests that co-induction with midazolam does not significantly alter the dosage of propofol required, as no significant differences were observed when compared to placebo [55]. 

#### 3.3.4. Sedation Protocol [32,52] (Table 2)

Induction should be conducted in a controlled manner, as the rapid administration of sedative agents may lead to respiratory instability and central sleep apnoea [49].

#### 3.3.5. Adjunctive Agents

To enhance patient comfort and optimise visualisation during endoscopy, various pre-procedural agents may be employed [56]. These include topical vasoconstrictors (e.g., oxymetazoline) [57], local anaesthetics (e.g., lidocaine), combination formulations (e.g., cophenylcaine), and adjunctive medications such as intravenous glycopyrrolate administered 30 min prior. However, we should be mindful that certain agents, such as atropine and excessive doses of lidocaine, may influence airway collapse patterns or alter sleep physiology. Anti-fogging agents could be applied to the endoscope for better view.

#### 3.3.6. Observation Window

Accurate assessment of the upper airway requires evaluation at an appropriate depth of sedation and during a stable respiratory pattern. If sedation is too light, airway obstruction may be underestimated compared to natural sleep; conversely, excessive sedation can exaggerate the degree of obstruction. Adequate sedation is characterised by the absence of arousal in response to vocal or tactile stimuli [49]. Furthermore, the patient should have undergone two to three cycles of airway obstruction, typically marked by stable snoring, hypoxic episodes, and apnoeic events [32,56]. The use of BIS monitoring has been proposed to guide sedation depth, with recommended values generally ranging between 50 and 80 [32,34,56]. 

#### 3.3.7. Procedure and Additional Manoeuvres

Upon initiation of sedation, the room should be dimmed to replicate the typical sleeping environment. Endoscopy should be performed to visualise the entire upper airway once the observation window is reached. Following airway assessment in the supine position, additional manoeuvres may be employed to enhance diagnostic insight.

A jaw thrust manoeuvre may be applied to identify suitable candidates for an OAT or mandibular advancement surgery [52,58]. However, a recent study suggests that the jaw thrust manoeuvre itself may have elicited arousal, thereby contributing to the observed improvement in upper airway obstruction. Among the 40 patients evaluated, 8 demonstrated an increase in BIS values exceeding 20 following the application of the manoeuvres [59].

To evaluate the anticipated therapeutic efficacy of OAT, a device advancing the mandible, such as a pre-existing OAT device, a simulation bite splint, or an adjustable boil-and-bite appliance, can be fitted onto the patients [60,61,62]. The upper airway may also be examined while the patient uses an existing PAP device to evaluate its effectiveness in maintaining airway patency [63,64]. Head rotation can be performed and has been utilised as a surrogate for simulating the lateral body position during DISE [65,66,67]. 

#### 3.3.8. Documentation

Over the years, various grading systems have been proposed to evaluate airway obstruction during DISE [68,69,70]. A 2017 systematic review identified fourteen studies that proposed different DISE classification systems [71]. This underscores the variability and subjectivity inherent in DISE reporting practices. Earlier frameworks included the employment of ordinal classifications, categorising collapses from simpler, single-level obstruction associated with less surgical necessity, to multi-level collapses that has higher surgical needs [72,73]. Classification was also proposed to provide predictive insights primarily in whether palatal surgery was recommended [74]. Sounds or noises were also considered in some of the classifications [75]. More recent approaches to DISE reporting have incorporated characterisation of the obstruction, including its site or level, collapse pattern, and severity. The sites include the soft palate, oropharynx, tonsils, tongue base, and epiglottis. Bilateral examination of the nasal cavity should be conducted to assess for enlarged turbinate, mucosal inflammation, or a deviated nasal septum, noting that these findings are consistent with those observed during awake examinations. The pattern and direction of obstructions should be documented, typically categorised as AP, lateral, or circumferential collapse (Figure 2a–c). Not all sites can exhibit every type of collapse. 

The severity of collapse is described in three degrees or quantified using percentage ranges. The degree of collapse can be classified as grade 0, no obstruction (no vibration); grade 1, partial obstruction (vibration); and grade 2, complete obstruction (collapse) [76,77,78]. Alternatively, percentage-based categorisation is generally divided into four tiers: 0–25% (grade 1), 25–50% (grade 2), 50–75% (grade 3), and 75–100% (grade 4) [79]. 

The VOTE classification system, established in 2011, has become a widely utilised tool for evaluating upper airway obstruction [23,80]. The acronym “VOTE” corresponds to the four anatomical regions of velum, oropharynx, tongue base, and epiglottis, providing a systematic framework for clinicians to assess site-specific patterns of obstruction. Among these regions, the velum can exhibit all three collapse types of AP, lateral, and concentric. In comparison, the oropharynx demonstrates only lateral collapse, the tongue base is restricted to AP collapse, and the epiglottis may present with either AP or lateral collapse. Another classification, called the nose oropharynx hypopharynx and larynx (NOHL) classification, was proposed to encompass all anatomic regions of the upper airway [79]. To promote consistency in DISE protocols and reporting, the Second European Position Consensus Meeting on DISE in 2017 reached an agreement to adopt the VOTE classification as the standardised reporting model, while allowing for supplementary annotations where appropriate [32]. However, a study using VOTE classification showed that within a sample size of 55, the interobserver consistency among three surgeons was good only at the level of oropharynx. The most diverse range of consistency was found in the diagnosis of velum obstruction [81]. To enhance inter-observer reliability, it would be beneficial to conduct calibration sessions across the workforce or among various observers, as well as to consider implementing dual-rating reviews in future studies.

In addition to the VOTE classification, the retropalatal narrowing may be further characterised as oblique, intermediate, or vertical, as proposed by Woodson [70]. These patterns correspond to the anatomical location of collapse: oblique narrowing occurs at the velum alone; intermediate involves both the velum and genu; and vertical encompasses the velum, genu, and hard palate. Moore’s classification can be used to describe four patterns of retroglossal narrowing, including type A (high tongue base), type B1 (high tongue base with retro-epiglottic narrowing), type B2 (diffuse tongue-base narrowing), and type 3 (isolated retro-epiglottic narrowing) [82].

Findings from the additional manoeuvres may be systematically recorded, with potential therapeutic interventions incorporated into the final DISE report (Figure 3, Figure 4, Figure 5 and Figure 6).

## 4. Treatment Implications from DISE Findings (Table 3 and Table 4)

In addition to enabling direct visualisation of upper airway obstruction in induced sleep, DISE has become a valuable modality for predicting the potential efficacy of therapeutic interventions, including PAP therapy, mandibular advancement device (MAD) and surgical procedures. With the ongoing advancement and standardisation of DISE protocols, recent studies have increasingly explored the correlation between specific obstruction patterns and their corresponding optimal treatment strategies [63].

### 4.1. Implications in PAP Therapy

Although PAP therapy is effective, its overall impact is often compromised by poor patient adherence. A major limitation of PAP therapy lies in its potential to deliver excessively high pressures that lead to discomfort, or insufficient pressures that fail to adequately prevent upper airway collapse. In one study with a sample size of 30, comparison was made between DISE guided CPAP (DISE-CPAP) to standard CPAP titration in patients with moderate to severe OSA [63]. The “optimum pressure—best possible” (complete or incomplete resolution of collapse) during DISE-CPAP was compared to the 95th percentile CPAP titration pressure. The study demonstrates that DISE-CPAP aids in determining optimal therapeutic pressure levels and identifying sites of airway collapse that are unresponsive to CPAP. The 95th percentile CPAP titration pressure frequently results in only partial resolution of airway patency. Patients with velum or oropharynx collapse are good CPAP candidates, while those with tongue base or epiglottic collapse are more resistant and may require surgery to optimise CPAP usage. This finding echoes to previous studies showing that epiglottic collapse acts as a major factor for intolerance to CPAP [37,83,84]. The higher VOTE score at the epiglottis level could also contribute to worse PAP adherence [85]. Another study also found that jaw laxity and associated mandibular retrusion at sleep onset was found to result in a complete tongue base obstruction. This form of obstruction was resistant to PAP therapy, resulting in an elevation of AHI, and DISE can assist in tackling the residual AHI in this subset of patients [86].

### 4.2. Implications in Positional Therapy

It is important to recognise that head rotation alone may not fully replicate the effects of a complete lateral body posture, which could influence the assessment of suitability of positional therapy. Improvement of the airway at the lateral head position was found in patients in positional obstructive sleep apnoea (POSA) [65]. A study indicated that head rotation in supine position yields similar sites, severity, and patterns of collapse to those observed in the lateral head and trunk position, with the exception at the level of the velum. Notably, AP collapse appears less pronounced during isolated head rotation than in full lateral positioning in one study [66]. However, another study demonstrated that differences were observed at all levels of obstruction when comparing effects of head rotation alone and full lateral body positioning in POSA patients. In non-positional OSA (NPOSA) patients, lateral head rotation and combined head and trunk rotation yield similar airway findings, except at the oropharynx [67]. CCC was observed less frequently among patients with POSA. Notably, the efficacy of upper airway surgical interventions was lower in POSA patients compared to those with NPOSA, but no significant differences were identified between the two groups in terms of treatment outcomes with MAD [87].

### 4.3. Implications in Mandibular Advancement Devices (MADs)

The optimal protocol during DISE to facilitate the identification of the therapeutic mandibular position and to predict treatment responders for MAD remains an area of ongoing investigation. A range of techniques may be employed based on resource availability, including jaw thrust, use of, patient’s existing OAT devices, simulation bite splints or adjustable boil-and-bite appliances.

An earlier study showed that patients demonstrating airway improvement in response to a 4 to 5 mm jaw thrust under DISE were more likely to respond favourably to MADs [88]. This observation aligns with findings from a separate study, which noted that patients exhibiting increased airway dimensions at the level of the velum and/or oropharynx in response to jaw thrust manoeuvres are likely to derive the greatest benefit from OAT [89]. However, a comparative study during DISE revealed only slight to moderate agreement between the jaw thrust manoeuvres and MADs in terms of the degree and configuration of airway obstruction. Notably, the poorest agreement was observed at the epiglottic level. While jaw thrust demonstrated a more superior improvement at the hypopharyngeal level, it was comparatively less effective at the retropalatal region [59]. This finding was reinforced by a separate study, which demonstrated no significant association between treatment response to MADs and pharyngeal expansion achieved via chin lift. In contrast, improvements in airway dimensions observed through DISE with MADs in the same patients revealed that responders exhibited greater retroglossal expansion [90]. The recent literature has raised concerns regarding the accuracy in the utility of manual jaw thrust in guiding treatment decisions. The manoeuvres per se can provoke arousal [59], and it lacks titratability and reproducibility both antero-posteriorly and vertically.

While titrating the patient’s existing OAT during DISE is feasible, using a simulation bite prior to device fabrication may be more clinically sound and applicable. The simulation bite can be produced via various methods. The patient’s maximal comfortable protrusion (MCP) can be identified and used to fabricating a removable Herbst appliance for use during DISE [61]. Alternatively, another study proposed using a dedicated registration fork to capture the MCP, wherein the upper arch was stabilised using bite registration material and cured, followed by fixation of the lower arch at the averaged MCP position [62]. One study utilised an adjustable thermoplastic boil-and-bite MAD, comprising two separate trays, and titrated to MCP in clinical setting, which was then used during DISE [60]. Collectively, they demonstrated an initial positive correlation between the simulation bite’s effect on upper airway patency during and the therapeutic response to MAD treatment. 

In a single-centre retrospective analysis involving 200 OSA who underwent DISE, manual jaw thrust resulted in a 66.7% reduction in the sum VOTE score. Lateral head rotation produced a reduction of 33.3% in patients with NPOSA and 50% in those with POSA. When these manoeuvres were combined, over 75% reduction was observed across all patient groups. These findings underscore the importance of individualised treatment planning, particularly for patients who may benefit from multimodal therapeutic strategies [91].

### 4.4. Implications in Palatal Surgeries

Multiple studies indicate that the use of DISE, as opposed to awake examination, significantly influences surgical decision-making in patients with OSA [37,92,93]. When interpreting findings across studies, it is important to recognise the diversity and ongoing evolution of surgical approaches performed in palatal procedures for OSA. This may also explain the reason for conflicting results into whether DISE improves surgical outcomes. It was found that presence of severe lateral pharyngeal wall and/or supraglottic collapse was associated with surgical failure, which included a variety of nasal and soft tissue surgeries [94]. A study of 20 patients showed that a significantly higher operative success rate was found in the group with retropalatal and tonsillar obstruction than the group with retroglossal and retro-epiglottic obstruction [95]. A multicentre study of 275 participants demonstrated that obstruction involving the oropharyngeal lateral walls and complete tongue base collapse was associated with poorer surgical outcomes [96]. Analysis of palatal shape revealed that increased AP narrowing at the genu was associated with poorer surgical outcomes, whereas greater lateral narrowing correlated with more favourable responses following isolated soft palate surgery [97]. 

However, a multicentre study of 326 OSA patients showed that the study group without DISE prior to nose, palatal, and/or tongue surgery showed better outcomes in terms of mean percentage reduction in AHI, mean increase in lowest oxygen saturation (LSAT), and mean blood pressure improvements [98]. In summary, studies that reported differences in surgical outcomes with the implementation of DISE identified oropharyngeal lateral wall collapse, as well as supraglottic and epiglottic obstruction, as predictors of poorer postoperative results.

### 4.5. Implications in Tongue Base Surgeries

Overall, the evidence supporting a significant impact of DISE on treatment success for tongue base surgeries remains limited. One study showed that despite poor agreement between Muller’s manoeuvre and DISE, superiority of DISE over Muller’s manoeuvre was not established [99]. Another investigation found that patients without oropharyngeal lateral wall collapse in DISE findings were more likely to experience postoperative improvement; however, this association did not translate to surgical success or cure [100].

### 4.6. Implications in Upper Airway Stimulation (UAS) Surgery 

The first implanted UAS device received FDA clearance in 2014 for the treatment of moderate to severe OSA, following the outcomes of the Stimulation Therapy for Apnoea Reduction (STAR) trial [101]. Given the novelty and associated cost of this therapeutic modality, ongoing research has focused on identifying patient populations most likely to benefit from UAS. Improved UAS outcomes were found in patients without palatal CCC during DISE [102]. A more recent multicentre study provided a more extensive analysis of DISE findings to AHI reduction and response rates to UAS, defined as a ≥50% reduction in AHI to fewer than 15 events per hour. Among different sites of obstruction, tongue-related collapse showed the strongest association with favourable response to UAS. In contrast, obstructions involving the oropharyngeal lateral walls and epiglottis were linked to less favourable outcomes. Although earlier studies identified palatal CCC as a contraindication for UAS, the authors emphasised that their findings did not demonstrate a statistically significant association between velum CCC and treatment outcomes. They suggested that previous conclusions may have been influenced by confounding factors [103].

### 4.7. Implications in Maxillomandibular Advancement (MMA)

Velum CCC and oropharyngeal lateral wall collapse represent the most challenging anatomical obstructions in the treatment of OSA for soft tissues surgeries. MMA has demonstrated comparable efficacy in reducing the AHI among patients with and without palatal CCC and has even been shown to resolve palatal CCC [104]. One study reported that the most pronounced post-MMA change in airway collapse was at the level of LPW. Patients who exhibited the greatest improvement in LPW collapsibility had the greatest reductions in AHI [105]. Similar authors conducted another study integrating DISE with computational fluid dynamics to evaluate dynamic airway and airflow changes following MMA demonstrated that treatment success, measured by improvements in AHI and oxygen desaturation index, was most strongly associated with reduced retropalatal airflow velocity as modelled by computational fluid dynamics, and enhanced LPW stability, as assessed by VOTE scoring during DISE [106]. To date, no studies have examined DISE findings and treatment outcomes in relation to variations in the magnitude, vector, or surgical modifications of MMA. The ideal jaw position after MMA should optimise airway outcomes, occlusion, and facial aesthetics. It will be interesting to compare the post-operative DISE changes with different MMA techniques. Theoretically, distraction osteogenesis can provide the most advancement in airway, but a study has shown that distraction has a higher complication rate and was not superior to sagittal split in improving the airway [107]. Future research should focus on leveraging DISE to identify patient-specific predictors of success across different MMA techniques, including segmental osteotomies designed to preserve facial aesthetics while optimising posterior airway expansion, particularly in individuals with Class I occlusion or bimaxillary protrusion [108].

**Table 3 diagnostics-15-02614-t003:** Summary table of key studies on various interventions with their respective design, DISE findings, and outcomes.

Intervention	Reference, Year	Design	Sample	DISE Patterns	Outcomes
PAP	[63], 2023	DISE-CPAP vs. standard CPAP titration 95th percentile pressure of the CPAP titration trial was compared to optimal pressure for alleviating upper airway blockage in DISE	30 (mean age 37.5, 17% women; moderate-to-severe OSA)	VOTE classification V & O: >80% had complete collapse	Mean optimal pressure to partially/fully open airway (DISE-CPAP): 16.1 ± 3.9 cmH_2_O Mean 95th percentile CPAP titration pressure: 14.3 ± 3.5 cmH_2_O CPAP often results in partial resolution of airway patency
Positional therapy	[87], 2018	Comparative POSA vs. NPOSA vs. non-OSA	860 DISE performed in 543 patients (119 non-OSA, 257 POSA, 167 NPOSA)	Velum and oropharynx collapse significantly determined presence of OSA CCC occurred less frequently in POSA compared to NPOSA	UAS often cured or improved OSA to less severe POSA Lower efficacy of upper airway surgical interventions in POSA compared to NPOSA
	[91], 2018	Retrospective, single-centre cohort study DISE with jaw thrust and lateral head rotation	200 (80.5% male; mean age 50.1 ± 11.7 years; BMI 27.0 ± 3.1 kg/m^2^; median AHI 19.2)	VOTE classification 44% non-positional, 56% positional (34% supine isolated, 66% supine predominant)	Jaw thrust reduced sum VOTE score by 66.7% across all subgroups Lateral head rotation reduced sum VOTE score by 33.3% in non-positional and supine predominant positional, and 50% in supine isolated positional Combined manoeuvres reduced sum VOTE score by >75% in all patients
	[66], 2015	Comparative Supine position with head rotated vs. head and trunk in lateral position	60 (44 male, mean AHI 20.8 ± 17.5)	VOTE classification Similar sites, severity, and patterns of collapse in head rotation (supine) vs. lateral head/trunk position, except at velum Head rotation: 15.0% and 15.0% had complete and partial AP velum collapse Full lateral position: 6.7% and 3.3% had complete and partial AP velum collapse	Head rotation yields similar DISE findings to full lateral positioning, except at velum level
MAD	[62], 2013	Observational Association between DISE findings with simulation bite and MAD treatment outcome	200 (74% male; age 46 ± 9 years; AHI 19 ± 13; BMI 27 ± 4 kg/m^2^) (135 had MAD)	Effects on upper airway patency with simulation bite	Positive correlation between simulation bite effects and MAD therapeutic response
	[88], 2007	Prospective Sleep nasoendoscopy with manual mandibular advancement and MAD outcome	120 (107/120 completed therapy, 8/107 dropped out)	Croft and Pringle scale Obstruction grades 3–5 with airway improvement with 4–5 mm jaw thrust	Favourable response to MAD in patients with airway improvement Median AHI reduced from 18.9 to 4.9 (*p* < 0.001); ESS score from 11 to 7 (*p* < 0.001)
	[61], 2020	Prospective cohort (single centre) Association between DISE findings with simulation bite and MAD treatment outcome	66 (median AHI 43.1)	Baseline collapses: Palate: 95.4% Oropharynx: 12.3% Tongue base: 61.5% Hypopharynx: 44.1% With simulation bite: Palate: 43% Oropharynx: 5.5% Tongue base: 18.7% Hypopharynx: 17.7% With chin lift: Palate: 30.2% Oropharynx: 1.1% Tongue base: 13% Hypopharynx: 4.4%	Presence of palatal collapse at baseline was associated with treatment response (OR: 8.6822; 95% CI: 1.5643–48.1894; *p*-value: 0.0135) In “predicted response” group, majority of patients were responders (83.3%)
	[60], 2020	Prospective cohort (single centre) Agreement in upper airway obstruction and configuration between jaw thrust and boil-and-bite MAD (MyTAP)	63	VOTE classification Degree of obstruction agreement (supine): V: 60% (*n* = 36, κ = 0.41), O: 68.3% (*n* = 41, κ = 0.35), T: 58.3% (*n* = 35, κ = 0.28), E: 56.7% (*n* = 34, κ = 0.14); (lateral): V: 81.7% (*n* = 49, κ = 0.32), O: 71.7% (*n* = 43, κ = 0.36), T: 90.0% (*n* = 54, κ = 0.23), E: 96.7% (*n* = 58, κ = not determined); Configuration agreement V: 69.0% (*n* = 20/29, κ = 0.41) in supine, 100% in lateral	Slight to moderate agreement in degree of obstruction between jaw thrust and boil-and-bite MAD Jaw thrust showed greater improvement in hypopharyngeal airway patency but less at retropalatal level
	90 [90], 2023	Comparative (chin lift vs. MAD)	56 (AHI ≥ 10 treated with MAD at 75% maximal protrusion)	Only laterolateral (LL) dimensions differed significantly with MAD presence at retro-epiglottic level, with significant relation of LL expansion ratio to treatment response (*p* = 0.0176)	Greater retroglossal expansion ratios in responders vs. non-responders (*p* = 0.0441) No significant association between response and chin lift
	[89], 2018	Retrospective MAD treatment outcome between DISE and no DISE	40 (20 DISE, 20 no DISE)	Increased airway dimensions at velum and/or oropharynx with jaw thrust	In DISE group: Significantly lower treatment AHI (*p* = 0.04), more patients reaching AHI < 5 with MAD (*p* = 0.04)
Palatal surgeries	[98], 2020	Prospective Nose, palate, and/or tongue surgery outcomes with and without DISE	326 (170 DISE, 156 no DISE; mean BMI 27.6 ± 4.6, 28.1 ± 3.9, mean AHI 15.9 ± 12.6, 13.2 ± 8.8)	VOTE classification	Without DISE: better outcomes (greater % AHI reduction, LSAT increase, blood pressure improvements) DISE may not significantly affect surgical success in OSA
	[96], 2019	Retrospective (multicentre) DISE findings and surgical outcomes	275 (14 centres) (mean age 51.4 ± 11.8, BMI 30.1 ± 5.2 kg/m^2^)	VOTE classification Most had velum obstruction and relatively few had epiglottis obstruction Primary structure obstructing: V: 35% (90 of 257) O: 24% (62 of 257) T: 39% (100 of 257) E: 2% (5 of 257)	Oropharyngeal lateral wall obstruction was associated with poorer surgical outcomes (adjusted odds ratio (AOR) 0.51; 95% CI 0.27, 0.93) Complete tongue-related obstruction was associated with lower odds of surgical response in moderate to severe OSA (AOR 0.52; 95% CI 0.28, 0.98) Surgical outcomes not clearly associated with degree and configuration of velum or the degree of epiglottis obstruction
	[97], 2024	Retrospective (multicentre) DISE assessed palatal shape and surgical outcomes of isolated pharyngeal surgery	209 (13 centres) (21% female; age 53.7 ± 11.5, BMI 30.3 ± 5.0 kg/m^2^)	Palatal levels: hard palate, genu AP, velum, and genu lateral Palatal shape was classified as vertical, intermediate, or oblique Vertical 4% Intermediate 54% Oblique 42%	Greater genu AP narrowing was associated with less odds of surgical response Greater genu lateral narrowing was associated with greater odds of surgical response Palate shape and other palate shape level scores were not clearly associated with surgical outcomes
	[94], 2012	Retrospective Identify DISE patterns as predictors of surgical failure	34	Severe airway obstruction was defined as >75%	Surgical failure group had severe lateral oropharyngeal wall collapse (73.3% vs. 36.8%, *p* = 0.037) and severe supraglottic collapse (93.3% vs. 63.2%, *p* = 0.046) as compared to surgical success group
	[95], 2015	Comparative	20 (16 male, age 19–57)	Upper airway classified as uvula and soft palate Lower airway classified as tongue base and epiglottis Modified Mallampati 1 & 2: Upper airway obstruction: 8/9 Lower airway obstruction: 2/9 Both: 1/9 Modified Mallampati 3 & 4: Upper airway obstruction: 9/11 Lower airway obstruction: 4/11 Both: 2/9	Significantly higher success rate in group with upper airway obstruction (*p* < 0.05) Significantly lower success rate was found in group with lower airway obstruction (*p* < 0.01)
Tongue base surgeries	[100], 2017	Retrospective DISE and predictive success for patients undergoing transoral robotic surgery	101	NOHL and VOTE classifications Success: Mean total VOTE score 3.3 ± 1.8 Mean total NOHL score 12.0 ± 1.9 No success: Mean total VOTE score 3.0 ± 1.9 Mean total NOHL score 11.8 ± 2.9	87% improvement, 51% success, 17% cured No oropharyngeal lateral collapse in VOTE was more likely to improve following surgery (*p* = 0.001); but effect not held for success or cure
	[99], 2020	Retrospective Muller’s manoeuvre (MM) vs. DISE on tongue base surgical outcomes	95 (47 MM, 48 DISE)	VOTE classification	Tonsil grade as significant predictive factor for surgical success in both groups (*p* = 0.004 in MM, *p* = 0.042 in DISE) Occlusion of the oropharyngeal lateral wall in MM group showed significant difference between surgical success and failure (*p* = 0.031), but not in DISE group (*p* = 0.596) Lack of evidence showing superiority of DISE over MM
UAS	[103], 2021	Retrospective cohort (multicentre) DISE and hypoglossal nerve stimulation outcomes	343 (10 centres, 76% male, age 60.4 ± 11.0, BMI 29.2 ± 3.6 kg/m^2^. AHI 35.6 ± 15.2)	Complete palate obstruction was associated with greatest AHI improvement (−26.8 ± 14.9) Complete tongue obstruction was associated with increased odds of treatment response (78%) Complete oropharyngeal lateral wall obstruction was associated with lower odds of surgical response (58%)	Primary tongue contribution to airway obstruction was associated with better outcomes Worse outcomes were found for oropharyngeal lateral walls obstruction CCC at velum should not be excluded for UAS
	[102], 2013	Observational DISE findings and UAS outcomes	21 (20 male, age 55 ± 11, BMI 28 ± 2 kg/m^2^, AHI 38.5 ± 11.8)	Levels classified as palate, oropharynx, tongue base, and epiglottis/hypopharynx Most common collapse patterns in this study were AP collapse at palate (76.2%) and the tongue base (71.4%)	Significantly better outcome without palatal CCC, reducing AHI from 37.6 ± 11.4 to 11.1 ± 12.0 (*p* < 0.001)
MMA	[106], 2016	Retrospective cohort study DISE and computational fluid dynamics (CFD) in evaluating post-MMA changes	20 (17 male, mean age 44 ± 12, BMI 27.4 ± 4.6 kg/m^2^, mean AHI 53.6 ± 26.6)	VOTE classification CFD modelling showed significant positive Pearson correlations between reduction of retropalatal airflow velocity and AHI (r = 0.617, *p* = 0.04)	AHI and ODI improvement post-MMA is best correlated with decreased retropalatal airflow velocity and increased lateral pharyngeal wall stability
	[105], 2015	Retrospective cohort DISE evaluation pre- and post-MMA	16 (15 male, average age 47 ± 10.9, BMI 29.4 ± 5.1 kg/m^2^, AHI 59.8 ± 25.6)	The post-MMA change in airway collapse was most significant at the lateral pharyngeal wall (*p* = 0.001) Subjects with most improvement in lateral pharyngeal wall had the largest changes in AHI (from 60.0 ± 25.6 to 7.5 ± 3.4) and oxygen desaturation index (ODI) (from 46.7 ± 29.8 to 5.3 ± 2; *p* = 0.002) Greatest reduction in upper airway collapsibility is seen at oropharynx, followed by velum, and tongue base	Stability of lateral pharyngeal wall is a marker of MMA success
	[104], 2020	Prospective case series DISE and MMA outcome, focusing on complete concentric collapse at level of palate	19 (14 with full dataset) (8 male, mean age 51 ± 7, BMI 25.6 ± 3.7 kg/m^2^, AHI 40.2 ± 25.6)	Levels classified as palate, oropharyngeal, tongue, hypopharyngeal and epiglottis 43% had CCC at palate All patients showed resolution of CCCp (*p* = 0.0159) during postoperative DISE	Comparable AHI reduction with or without CCC MMA resolved palatal CCC

**Table 4 diagnostics-15-02614-t004:** Summary table of DISE implications and limitations of various treatment modalities for OSA.

Treatment Modality	DISE Implications and Limitations
PAP therapy	Determines optimal therapeutic pressure Identifies CPAP-resistant collapse sites (tongue base and epiglottis) Higher VOTE score at epiglottis associated with worse adherence
Positional therapy	POSA shows airway improvement with lateral head position, except at velum level
Palatal surgeries	Jaw thrust-resultant airway patency improvements predict MAD response, especially at velum and oropharynx, yet lacks titratability and may cause arousal Simulation bites correlate with MAD success
Tongue base surgeries	Oropharyngeal lateral wall and epiglottic collapse predict poorer outcomes Retropalatal or tonsillar obstructions has higher success than retroglossal or retro-epiglottic obstructions Genu lateral narrowing predicts better outcomes Conflicting results on DISE’s impact on surgical success Limited evidence of DISE improving surgical success Absence of oropharyngeal lateral wall collapse linked to postoperative improvement, but not cure
UAS	Tongue-related collapse predicts favourable UAS response Oropharyngeal lateral wall and epiglottic collapse linked to poorer outcomes Palatal CCC utilised as contraindication to UAS
MMA	Effective for velum CCC and oropharyngeal lateral wall collapse No studies on variations in MMA magnitude/vector or surgical modifications

## 5. Conclusions

DISE has emerged as a critical diagnostic tool to optimise treatment planning for OSA patients. Its predictive accuracy is still debated due to methodological variability in the earlier evidence. Efforts have been made in unifying the protocol from sedation protocol to documentation via the VOTE classification, to serve as the basis on more standardised research on treatment applications. DISE offers critical insights that guide clinical decision making. It is recommended as a pre-operative diagnosis prior to surgical intervention, and serves as a valuable tool in evaluating refractory cases of previous treatment failure. Future research should focus on larger scale studies on associations between treatment success and different collapse patterns or additional manoeuvres used in DISE. Integration of findings of DISE in distinct OSA phenotypes and integrating tools like computational fluid dynamics should be considered to provide more insights into treatment outcome.

## Figures and Tables

**Figure 2 diagnostics-15-02614-f002:**
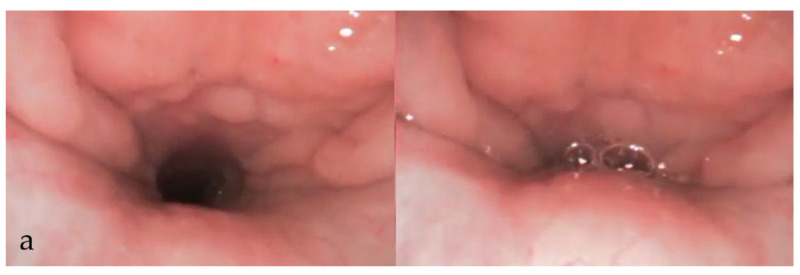
DISE images showing velum AP (**a**), lateral (**b**), and circumferential (**c**) collapse.

**Figure 3 diagnostics-15-02614-f003:**
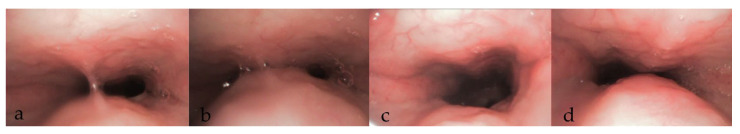
DISE images showing changes in collapse at velum (**a**,**b**) with jaw thrust (**c**) and mandibular advancement device (MAD) (**d**).

**Figure 4 diagnostics-15-02614-f004:**
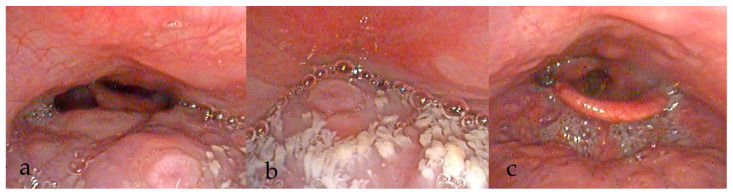
DISE images showing changes in collapse at tongue base (**a**,**b**) with jaw thrust (**c**).

**Figure 5 diagnostics-15-02614-f005:**
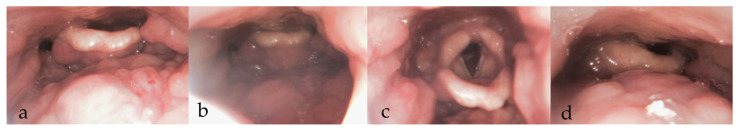
DISE images showing changes in collapse at epiglottis (**a**,**b**) with jaw thrust (**c**) and MAD (**d**).

**Figure 6 diagnostics-15-02614-f006:**
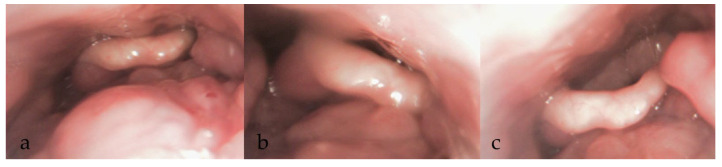
DISE images showing changes in collapse (**a**) with head positioning to left (**b**) and right (**c**).

**Table 1 diagnostics-15-02614-t001:** Advantages and disadvantages of various sedative drugs [32,34,35,50,51,52,53].

Sedative Drugs	Advantages	Disadvantages
Propofol	Rapid onset and shorter time to adequate sedation Effective titration with target-controlled infusion (TCI) Mimics NREM sleep Less muscle relaxation	Higher hypoxemia risk Dose-dependent increase in upper airway collapsibility and genioglossus muscle inhibition
Midazolam	Rapid onset Mimics NREM sleep stage 1 and 2 Antidote available	Respiratory depressant effects May increase sneezing
Dexmedetomidine	Preserves natural sleep architecture Lower hypoxemia risk Minimal respiratory depression	Long onset Chance of failed sedation

**Table 2 diagnostics-15-02614-t002:** Sedative protocol of different sedative drugs.

Sedative Drugs	Bolus	Target-Controlled Infusion
Propofol	Starting dose: 30–50 mg with increasing rate of 10 mg/2 min; or Starting dose: 1 mg/kg with increasing rate of 20 mg/2 min	Starting dose: 1.5–3.0 µg/mL then increasing rate 0.2–0.5 µg/mL/2 min
Midazolam	Starting dose: 0.03 mg/kg, increasing rate of 0.03 mg/kg after 2–5 min, 0.015 mg/kg after 5 min	/
Dexmedetomidine	Starting dose: 1.5 µg/kg over 10 min; maintenance: 1.5 µg/kg/h	Starting dose: 1 µg/kg for 10 min; then 1 µg/kg/h

## Data Availability

Not applicable.

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
