# Peer review of "Current Concepts of the Applications and Treatment Implications of Drug-Induced Sleep Endoscopy for the Management of Obstructive Sleep Apnoea"

_diagnostics, 2025, doi:10.3390/diagnostics15202614_

Round 1
Reviewer 1 Report
Comments and Suggestions for Authors
Thanks for the opportunity to review the manuscript. It is written in clinician-friendly and very practical language, and summarises all relevant aspects of DISE. Conclusions are appropriately cautious about inter-observer variability and sedation effects. I only have some minor recommendations for improvements on methodology and evidence synthesis:
- Add a brief "methods" section, even though it is a narrative review (which databases searched, date range, key terms and how evidence was selected / weighted). This will better anchor the strong practical content.
- Include a summary table of key studies per intervention (PAP/ DISE-PAP, MAD, palatal, tongue base, epiglottis, UAS, MMA): design, sample size, DISE patterns, primary outcomes and magnitude of effect. This will make the signal vs. noise visible and justify the cautious tone of your conclusions.
- Consider adding short guidance on how to improve Inter-observer reliability (dual-rater review, calibration sessions / trainings).
- You acknowledge anaesthesia risk but do not quantify adverse events. A short paragraph summarising typical DISE adverse events (hypoxemia events, airway maneuvers, aborted procedures) and recommending documentation of rescue steps would strengthen the “During DISE” section.
Author Response
Response to Reviewer 1
Comments 1: Add a brief "methods" section, even though it is a narrative review (which databases searched, date range, key terms and how evidence was selected / weighted). This will better anchor the strong practical content.
Response 1: Thank you for the comment. A “methods” section was added to the manuscript.
Comments 2: Include a summary table of key studies per intervention (PAP/ DISE-PAP, MAD, palatal, tongue base, epiglottis, UAS, MMA): design, sample size, DISE patterns, primary outcomes and magnitude of effect. This will make the signal vs. noise visible and justify the cautious tone of your conclusions.
Response 2: Thank you for the suggestion. A summary table has been included per your suggestions as Table 3.
Comments 3: Consider adding short guidance on how to improve Inter-observer reliability (dual-rater review, calibration sessions / trainings).
Response 3: Thank you for the suggestion. A short guidance has been added under “2.3.7. Documentation”.
Comments 4: You acknowledge anaesthesia risk but do not quantify adverse events. A short paragraph summarising typical DISE adverse events (hypoxemia events, airway maneuvers, aborted procedures) and recommending documentation of rescue steps would strengthen the “During DISE” section.
Response 4: Thank you for the comment. Two short paragraphs were incorporated under “2.2.3. Pre-anaesthetic consultation”, and “2.3. During DISE”.
Reviewer 2 Report
Comments and Suggestions for Authors
The manuscript is well crafted, with a clear structure and a pertinent, up-to-date literature review. The information provided is sufficient and coherent to support the conclusions, and the topic, the applications and treatment implications of DISE in obstructive sleep apnoea, is highly relevant and of great interest to the medical community. Overall, this work adds clinical value and may inform practice and decision-making.
Define article type and methodology. Please specify whether this is a narrative review, scoping review, or systematic review. If systematic/scoping, include search strategy (databases, dates, keywords, inclusion/exclusion), study selection flow (PRISMA), and risk-of-bias approach. If narrative, briefly describe how the literature was selected to minimize selection bias and justify the coverage.
The tables are accurate and readable. They include clear titles and footnotes and well-labeled columns
I recommend adding to the conclusion section a brief paragraph that states the study’s clinical relevance, and the clinical scenarios in which their application offers the greatest benefit.
Author Response
Response to Reviewer 2
Comments 1: Define article type and methodology. Please specify whether this is a narrative review, scoping review, or systematic review. If systematic/scoping, include search strategy (databases, dates, keywords, inclusion/exclusion), study selection flow (PRISMA), and risk-of-bias approach. If narrative, briefly describe how the literature was selected to minimize selection bias and justify the coverage.
Response 1: Thank you for the comment. A “methods” section was added to the manuscript.
Comments 2: I recommend adding to the conclusion section a brief paragraph that states the study’s clinical relevance, and the clinical scenarios in which their application offers the greatest benefit.
Response 2: Thank you for the suggestion. A brief paragraph was added to the conclusion which highlighted when should DISE be performed clinically.
Reviewer 3 Report
Comments and Suggestions for Authors
- The introduction is rather long and could be shortened to improve readability and maintain focus. I suggest condensing the general background on OSA while emphasizing contemporary approaches to its management. This will make the introduction more relevant to the objectives of your review. You may also consider citing recent work on the topic (https://doi.org/10.3390/oral5030055), https://doi.org/10.3390/life14121652 to strengthen the context and highlight current perspectives in OSA management.
2.The conclusion provides a good summary of the role of DISE in OSA management. However, it could be made more concise and impactful. At present, it reads like an extension of the discussion rather than a focused closing statement. I recommend shortening the text to highlight only the key take-home messages: (1) DISE is a valuable diagnostic tool that influences treatment planning, (2) its predictive accuracy is still debated due to methodological variability, and (3) future research should address standardization and integration with other diagnostic modalities. A more streamlined conclusion would leave the reader with a stronger final impression of your work.
3.Please ensure that all abbreviations are defined at first mention in the main text, even if they have already been defined in the abstract. For example, “OSA” is used in the introduction without being defined (it should be introduced as “obstructive sleep apnea (OSA)” at first mention). Similarly, review the entire manuscript to confirm that each abbreviations are properly defined before first use.
4.In addition, please carefully check the manuscript for typographical errors, punctuation, and spacing inconsistencies to improve readability and professionalism.
5.Please revise the keywords to ensure they are Medical Subject Headings (MeSH) terms and list them in alphabetical order. This will improve indexing and retrieval of your article.
Author Response
Response to Reviewer 3
Comments 1: The introduction is rather long and could be shortened to improve readability and maintain focus. I suggest condensing the general background on OSA while emphasizing contemporary approaches to its management. This will make the introduction more relevant to the objectives of your review. You may also consider citing recent work on the topic (https://doi.org/10.3390/oral5030055), https://doi.org/10.3390/life14121652 to strengthen the context and highlight current perspectives in OSA management.
Response 1: Thank you for the comment. Both publications have been included in the Introduction.
Comments 2: The conclusion provides a good summary of the role of DISE in OSA management. However, it could be made more concise and impactful. At present, it reads like an extension of the discussion rather than a focused closing statement. I recommend shortening the text to highlight only the key take-home messages: (1) DISE is a valuable diagnostic tool that influences treatment planning, (2) its predictive accuracy is still debated due to methodological variability, and (3) future research should address standardization and integration with other diagnostic modalities. A more streamlined conclusion would leave the reader with a stronger final impression of your work.
Response 2: Thank you for the comment. The conclusion has been amended with the consideration of comments from different reviewers.
Comments 3: Please ensure that all abbreviations are defined at first mention in the main text, even if they have already been defined in the abstract. For example, “OSA” is used in the introduction without being defined (it should be introduced as “obstructive sleep apnea (OSA)” at first mention). Similarly, review the entire manuscript to confirm that each abbreviations are properly defined before first use.
Response 3: Thank you for the reminder. The entire manuscript has been reviewed again and abbreviations have been defined before first use.
Comments 4: In addition, please carefully check the manuscript for typographical errors, punctuation, and spacing inconsistencies to improve readability and professionalism.
Response 4: Thank you for the comment. The manuscript has been reviewed again.
Comments 5: Please revise the keywords to ensure they are Medical Subject Headings (MeSH) terms and list them in alphabetical order. This will improve indexing and retrieval of your article.
Response 5: Thank you for the reminder. The keywords have been amended to MeSH terms and listed in alphabetical order.